# Fitness models provide accurate short-term forecasts of SARS-CoV-2 variant frequency

**Eslam Abousamra** [1,2]* *, **Marlin Figgins** [1,3]*, **Trevor Bedford** [1,2,4]

**1** Vaccine and Infectious Disease Division, Fred Hutchinson Cancer Center, Seattle, Washington, United States of America, **2** Department of Epidemiology, University of Washington, Seattle, Washington, United States of America, **3** Department of Applied Mathematics, University of Washington, Seattle, Washington, United States of America, **4** Howard Hughes Medical Institute, Seattle, Washington, United States of America

* These authors contributed equally to this work.
* eabousam@uw.edu

## Abstract

Genomic surveillance of pathogen evolution is essential for public health response, treatment strategies, and vaccine development. In the context of SARS-COV-2, multiple models have been developed including Multinomial Logistic Regression (MLR) describing variant frequency growth as well as Fixed Growth Advantage (FGA), Growth Advantage Random Walk (GARW) and Piantham parameterizations describing variant $R_t$. These models provide estimates of variant fitness and can be used to forecast changes in variant frequency. We introduce a framework for evaluating real-time forecasts of variant frequencies, and apply this framework to the evolution of SARS-CoV-2 during 2022 in which multiple new viral variants emerged and rapidly spread through the population. We compare models across representative countries with different intensities of genomic surveillance. Retrospective assessment of model accuracy highlights that most models of variant frequency perform well and are able to produce reasonable forecasts. We find that the simple MLR model provides ∼0.6% median absolute error and ∼6% mean absolute error when forecasting 30 days out for countries with robust genomic surveillance. We investigate impacts of sequence quantity and quality across countries on forecast accuracy and conduct systematic downsampling to identify that 1000 sequences per week is fully sufficient for accurate short-term forecasts. We conclude that fitness models represent a useful prognostic tool for short-term evolutionary forecasting.

## Author summary

Over the course of the COVID-19 pandemic, SARS-CoV-2 evolved into many different genetic variants such as the well known Alpha, Beta, Gamma and Delta variants in early 2021 and the Omicron variant in late 2021. These genetic variants could more easily spread from person to person and so outcompeted previous versions of the virus. Even if they aren't being given Greek letter names, new variants are still arising with recent waves of COVID-19 caused by variants such as XBB and JN.1. Predicting which variants will increase in frequency and which variants will decrease in frequency is important for

**Data Availability Statement:** Sequence data including date and location of collection as well as clade annotation was obtained via the Nextstrain-curated data set that pulls data from GISAID database. A full list of sequences analyzed with

accession numbers, derived data of sequence counts and case counts, along with all source code used to analyze this data and produce figures is available via the GitHub repository github.com/blab/ncov-forecasting-fit.

**Funding:** MF is an ARCS Foundation scholar and was supported by the National Science Foundation Graduate Research Fellowship Program under Grant No.\ DGE-1762114. https://arcsfoundation.org/. TB is an Investigator of the Howard Hughes Medical Institute and employee of the Howard Hughes Medical Institute. https://www.hhmi.org/. This work is supported by NIH NIGMS R35 GM119774 awarded to TB and by a HHMI COVID-19 Collaboration Initiative awarded to TB. The funders had no role in study design, data collection and analysis, decision to publish, or preparation of the manuscript.

**Competing interests:** The authors have declared that no competing interests exist.

public health, particularly in terms of updating the formulation of the annual COVID-19 vaccine. In this paper, we investigate statistical models that use observed frequencies of different variants in the past weeks to estimate the frequency of different variants today and to forecast the frequency of different variants in 30 days time. We find that in countries with sufficient amounts and timeliness of genetic sequence data, that models forecast well and can be a useful tool for public health.

## Introduction

The emergence of acute respiratory virus SARS-CoV-2 causing COVID-19 disease and its subsequent circulating variants severely impacted global health and worldwide economies [1]. Due to its rapid evolution, original SARS-CoV-2 strains were replaced by derived, selectively advantageous variant lineages during 2021 [2], with Omicron, a highly transmissible and immune evasive variant becoming the dominant strain in early 2022 [3]. It has become increasingly evident that monitoring the evolution and dissemination of these variants remains crucial with SARS-CoV-2 continuing to evolve beyond Omicron [4]. Forecasting variant dynamics allows us to make informed decisions about vaccines and to predict variant-driven epidemics.

Fitness models are a key framework for forecasting changes in variant frequency through time. These models were first introduced for the study of seasonal influenza virus [5–7] and there have relied on correlates of viral fitness such as mutations to epitope sites on influenza's surface proteins. In modeling emergence and spread of SARS-CoV-2 variant viruses, the use of Multinomial Logistic Regression (MLR) has become commonplace [8–11]. Here, MLR is analogous to a population genetics model of a haploid population in which different variants have a fixed growth advantage and are undergoing Malthusian growth. As such, it presents a natural model for describing evolution and spread of SARS-CoV-2 variants. Additionally, models introduced by Figgins and Bedford [12] incorporate case counts and variant-specific $R_t$, but still can be used to project variant frequencies while Piantham et al [13] does not incorporate them.

Here, we systematically assess the predictive accuracy of fitness models for nowcasts and short-term forecasts of SARS-CoV-2 variant frequencies. We focus on variant dynamics during 2022 in which multiple sub-lineages of Omicron including BA.2, BA.5 and BQ.1 spread rapidly throughout the world. We compare across several countries including Australia, Brazil, Japan, South Africa, Trinidad and Tobago, the United Kingdom, the United States, and Vietnam to assess genomic surveillance systems with different levels of throughput and timeliness. To assess the performance of these models, we used mean and median absolute error (AE) as a metric to compare the predicted frequencies to retrospective truth. This metric allowed us to evaluate the accuracy and reliability of the models and to identify those that were most effective in predicting SARS-CoV-2 variant frequency. We also examined aspects of country-level genomic surveillance that contribute to errors in these models and explored the role of sequence availability on nowcast and forecast errors through downsampling sequencing efforts.

## Results

### Reconstructing real-time forecasts

We focus on SARS-CoV-2 sequence data shared to the GISAID EpiCoV database [14]. Each sequence is annotated with both a collection date, as well as a submission date. We seek to reconstruct data sets that were actually available on particular 'analysis dates', and so we use use submission date to filter to sequences that were available at a specific analysis date. We

additionally filter to sequences with collection dates up to 90 days before the analysis date. We categorize each sequence by Nextstrain clade (21K, 21L, etc. . .) as such clades are generally at a reasonable level of granularity for understanding adaptive dynamics [15]; there are 7 clades circulating during 2022 vs hundreds of Pango lineages. Resulting data sets for representative countries Japan and the USA for analysis dates of Apr 1 2022, Jun 1 2022, Sep 1 2022 and Dec 1 2022 are shown in Fig 1A, while S1 Fig shows data sets for Australia, Brazil, South Africa, Trinidad and Tobago, the UK, and Vietnam. We see consequential backfill in which genome sequences are not immediately available and instead available after a delay due to the necessary bottlenecks of sample acquisition, testing, sequencing, assembly and data deposition. Thus, even estimating variant frequencies on the analysis date as a nowcast requires extrapolating from past week's data. Different countries with different genomic surveillance systems have different levels of throughput as well as different amounts of delay between sample collection and sequence submission [16].

We employ a sliding window approach in which we conduct an analysis twice each month (on the 1st and the 15th) and estimate variant frequencies from −90 days to +30 days relative to each analysis date. We illustrate our frequency predictions using the MLR model showing resulting trajectories for Japan and the US in Fig 1B and showing trajectories for Australia, Brazil, South Africa, Trinidad and Tobago, the UK, and Vietnam in S2–S7 Figs. Sometimes we see initial over-shoot or under-shoot of variant growth and decline, but there is general consistency across trajectories. Additionally, we retrospectively reconstructed the simple 7-day smoothed frequency across variants and present these trajectories as solid black lines. We treat this retrospective trajectory as 'truth' and thus deviations from model projections and retrospective truth can be assessed to determine nowcast and short-term forecast accuracy. Consistent with less available data, we observe that the model predictions for Japan were more frequently misestimated compared to the United States with particularly large differences for clades 22B (lineage BA.5) and 22E (lineage BQ.1) (Fig 1B).

## Model error comparison

We utilize five models for predicting the frequencies of SARS-CoV-2 variants. The simplest of these models is Multinomial Logistic Regression (MLR) commonly used in SARS-CoV-2 analyses [8–11], which uses only variant-specific sequence counts and has a fixed growth advantage for each variant. More complex models include the Fixed Growth Advantage (FGA) and Growth Advantage Random Walk (GARW) parameterizations of the variant $R_t$ model introduced by Figgins and Bedford [12], which uses case counts in addition to variant-specific sequence counts. The Piantham et al. model [13] operates on a similar principle in estimating relative fitness, but differs in model details and does not use case counts. We compare these four models to a naive model to serve as a reference for comparison. The naive model is implemented as a 7-day moving average on the retrospective raw frequencies using the most recent seven days for which sequencing data is available. We compare forecasting accuracy across different time lags from −30 days back from date of analysis as hindcast, to +0 days from date of analysis as nowcast, and +30 days forward from date of analysis as forecast.

We refer to the absolute error $\mathrm{AE}_t^{m,d}$ for a given model $m$, data set $d$ and time $t$ as the difference between the retrospective 7-day smoothed frequency and the model predicted frequency (see Methods). We calculate median absolute error and mean absolute error across datasets and across time lags to assess the relative performance of the models for the eight countries (Fig 2 and Table 1). As expected, we observe decreasing performance across models as lags increase from −30 days to +30 days. For example, median absolute error increases for the MLR model from 0.1–1.4% at −30 days, to 0.3–2.0% at 0 days and to 0.5–1.9% at +30 days. Similarly,

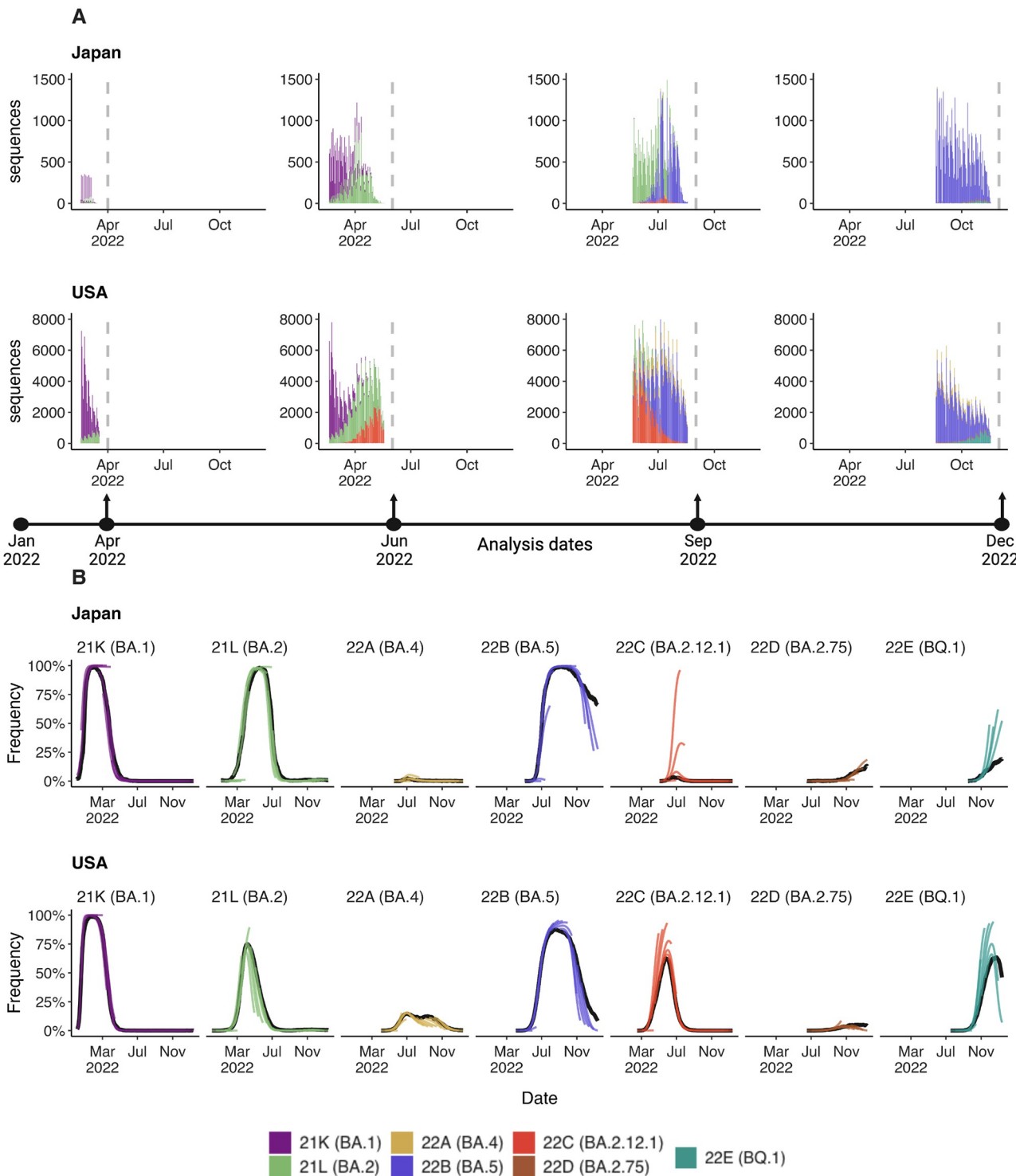

**Fig 1. Reconstructing available data sets and corresponding predictions for Japan and USA.** (A) Variant sequence counts categorized by Nextstrain clade from Japan and United States at 4 different analysis dates. (B) +30 day frequency forecasts for variants in bimonthly intervals using the MLR model. Each forecast trajectory is shown as a different colored line. Retrospective smoothed frequency is shown as a thick black line.

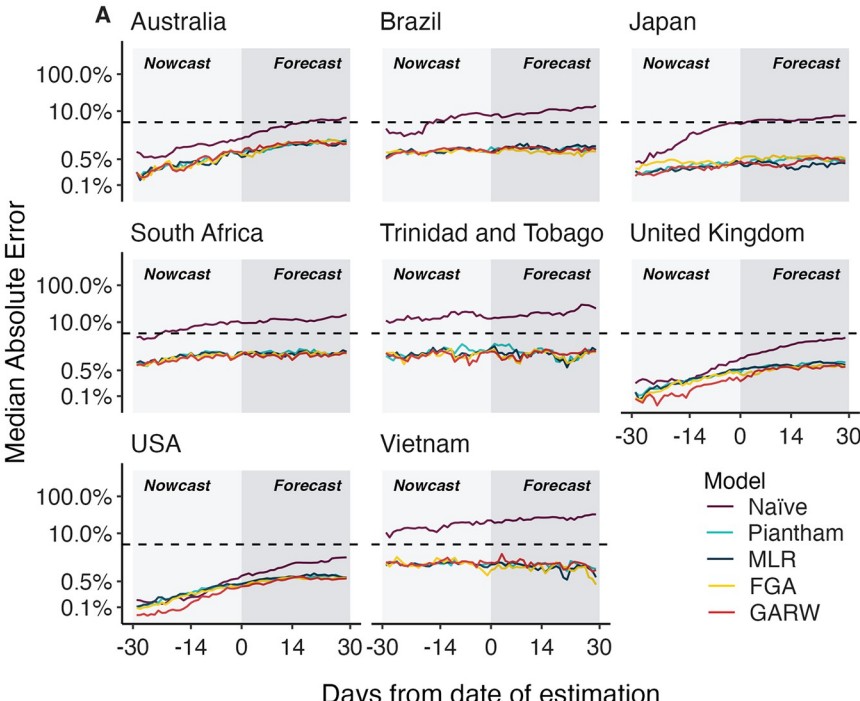

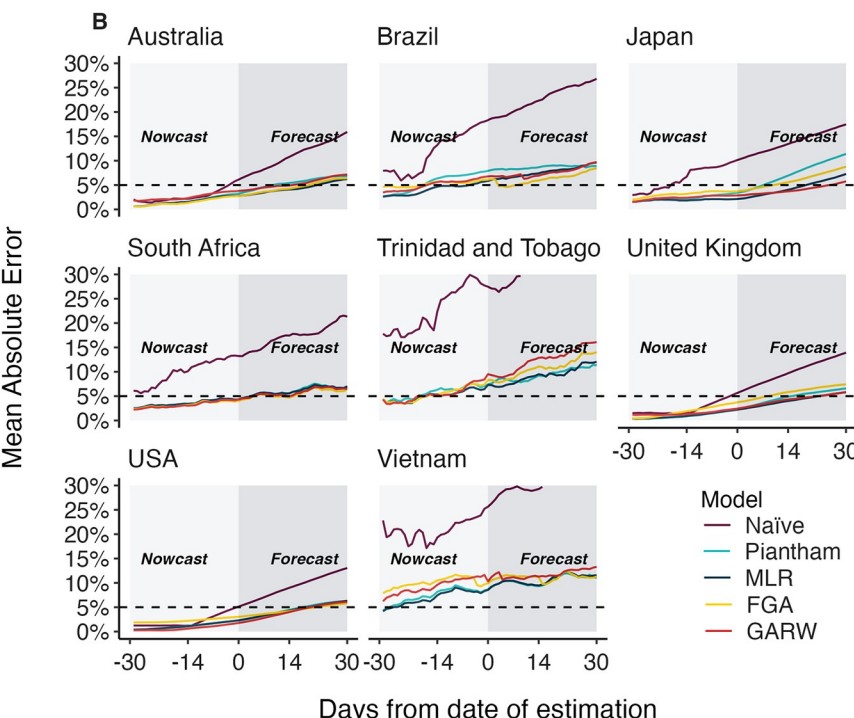

**Fig 2. Absolute error across models, countries and forecast lags.** (A) Median absolute error and (B) mean absolute error across countries, models and forecast lags moving from −30 day hindcasts to +30 day forecasts. For each county / model / lag combination, the median and the mean are summarized across analysis data sets. Panel A uses a log y axis for legibility while panel B uses a natural y axis.

**Table 1. Median and mean absolute error across models, countries and forecast lags.** Models with the lowest error for each country / lag combination are bolded for clarity.

| Location | Median Absolute Error | | | | | Mean Absolute Error | | | | |
|---|---|---|---|---|---|---|---|---|---|---|
| | Naïve | Piantham | MLR | FGA | GARW | Naive | Piantham | MLR | FGA | GARW |
| **-30 Lead from date of estimation** | | | | | | | | | | |
| Australia | 0.80% | **0.20%** | **0.20%** | **0.20%** | **0.20%** | 2.10% | **0.60%** | **0.60%** | **0.60%** | 1.80% |
| Brazil | 3.50% | 0.80% | 0.70% | 0.80% | **0.60%** | 7.60% | 2.50% | **2.40%** | 4.60% | 3.30% |
| Japan | 0.40% | **0.20%** | **0.20%** | **0.20%** | **0.20%** | 2.90% | **1.40%** | **1.40%** | 1.90% | **1.40%** |
| South Africa | 3.70% | 1.00% | 0.90% | **0.80%** | **0.80%** | 5.50% | 2.30% | 2.50% | **2.20%** | **2.20%** |
| Trinidad and Tobago | 12.50% | 1.50% | **1.40%** | **1.40%** | **1.40%** | 19.90% | **4.20%** | **4.20%** | **4.20%** | **4.20%** |
| USA | 0.20% | **0.10%** | **0.10%** | **0.10%** | **0.10%** | 1.30% | 0.40% | 0.40% | 1.80% | **0.20%** |
| United Kingdom | 0.20% | **0.10%** | **0.10%** | **0.10%** | **0.10%** | 1.50% | 0.40% | 0.40% | 0.50% | 1.20% |
| Vietnam | 10.40% | 1.50% | **1.30%** | 1.40% | 1.40% | 21.00% | **4.00%** | **4.00%** | 7.80% | 6.20% |
| **0 Lead from date of estimation** | | | | | | | | | | |
| Australia | 1.80% | 0.80% | **0.60%** | **0.60%** | 0.70% | 6.10% | 3.20% | **2.80%** | **2.70%** | 3.80% |
| Brazil | 7.20% | 1.10% | 1.00% | **0.90%** | 1.00% | 18.30% | 7.90% | **5.90%** | **6.10%** | 6.80% |
| Japan | 4.50% | 0.50% | **0.30%** | 0.50% | 0.40% | 10.10% | 3.40% | **2.10%** | 3.70% | **2.90%** |
| South Africa | 9.30% | 1.50% | 1.60% | 1.50% | **1.30%** | 13.20% | **4.30%** | **4.30%** | 4.00% | **4.30%** |
| Trinidad and Tobago | 12.50% | 1.90% | 2.00% | 1.70% | **1.60%** | 27.50% | 7.40% | 7.30% | 8.30% | 9.50% |
| USA | 0.60% | **0.40%** | **0.40%** | **0.40%** | **0.40%** | 5.10% | **2.30%** | **2.30%** | 3.00% | **1.80%** |
| United Kingdom | 1.10% | 0.50% | 0.50% | 0.40% | **0.20%** | 5.70% | **2.30%** | **2.20%** | 3.70% | **2.40%** |
| Vietnam | 22.30% | 1.50% | 1.40% | **1.20%** | 1.70% | 25.60% | 8.70% | **8.60%** | 9.90% | 10.30% |
| **30 Lead from date of estimation** | | | | | | | | | | |
| Australia | 6.20% | 1.60% | 1.50% | 1.50% | **1.40%** | 15.90% | 6.80% | **6.20%** | 6.40% | 7.10% |
| Brazil | 13.40% | **0.90%** | 1.20% | 1.00% | 1.20% | 26.80% | 8.90% | **8.40%** | 9.70% | |
| Japan | 7.50% | **0.50%** | **0.50%** | **0.50%** | **0.50%** | 17.50% | 11.40% | 7.30% | 8.80% | **5.80%** |
| South Africa | 15.80% | 1.50% | 1.60% | 1.60% | **1.40%** | 21.40% | 6.90% | 7.00% | **6.40%** | 6.50% |
| Trinidad and Tobago | 23.50% | 2.00% | 1.90% | 1.60% | **1.30%** | 38.60% | **11.30%** | 12.00% | 14.00% | 16.10% |
| USA | 2.00% | **0.60%** | 0.70% | **0.60%** | **0.60%** | 13.10% | 6.30% | 6.30% | **5.80%** | 6.10% |
| United Kingdom | 3.60% | 0.80% | 0.70% | **0.60%** | **0.60%** | 13.90% | 6.60% | **5.80%** | 7.40% | **5.80%** |
| Vietnam | 32.10% | 1.60% | 1.10% | **0.80%** | 1.10% | 33.20% | **11.00%** | 11.60% | 11.30% | 13.30% |

mean absolute error increases for the MLR model from 0.4–4.2% at −30 days, to 2.2–8.6% at 0 days and to 5.8–12.0% at +30 days. All four forecasting models perform better than the naive model, with all four models exhibiting similar performance. We observe a larger decrease in performance as lags increase in terms of mean absolute error compared to median absolute error. Absolute error varies substantially across predictions for individual analysis dates and variants with most predictions having very little error, while a subset of predictions have larger error (Fig 3). This skewed distribution results in the large observed differences between median and mean summary statistics. Thus, models predict frequencies well most of the time, but are occasionally incorrect and the proportion of incorrect predictions increases through time.

In addition to calculating median and mean absolute error, we estimate the coverage of 95% posterior latent frequencies (S8(A) Fig) and posterior predictive sample frequencies (S8 (B) Fig) across models. We generate the posterior predictive coverage by sampling random counts for each variant using their posterior latent frequencies conditioning on the total sequences being those observed retrospectively. We find that the posterior predictive coverage is generally higher and a better fit for the models in question. Additionally, we find that the coverage is lower in countries with the highest sequencing intensity like the US and UK,

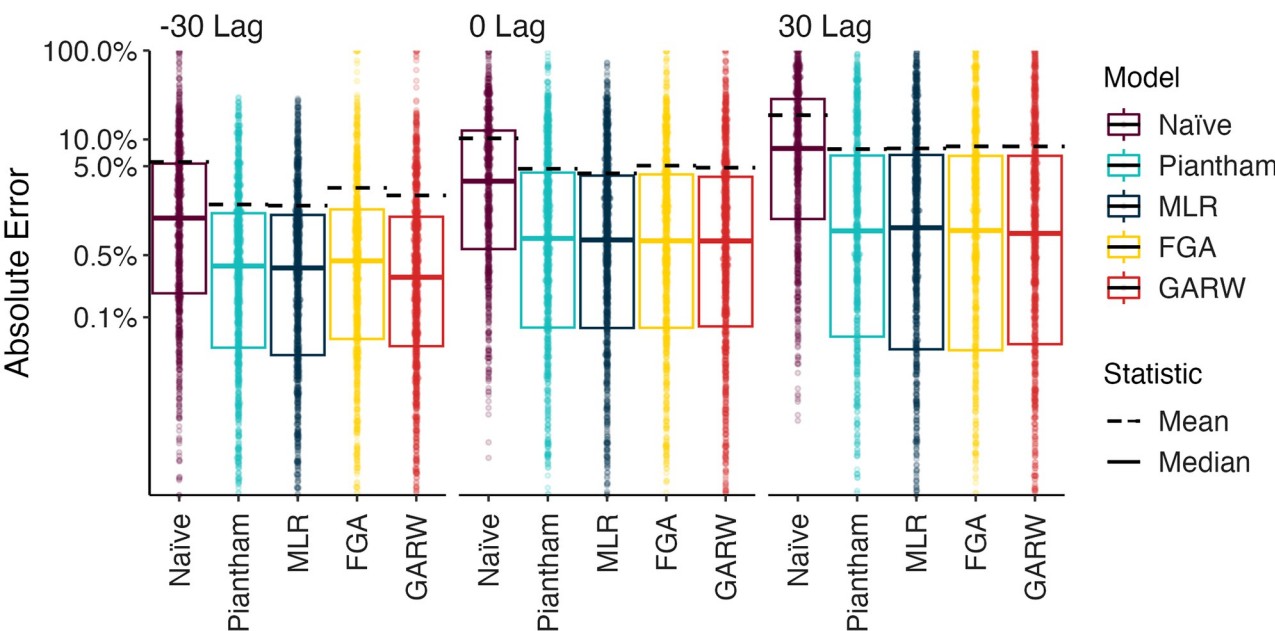

**Fig 3. Absolute error across models, countries and forecast lags.** Distribution of absolute error on a log scale across models and across forecast lags. Each point represents the absolute error for a data set / country combination. Solid lines show the median of these distributions and dashed lines show the means of these distributions.

suggesting that there may be over-dispersion in the sequence counts relative to binomial or multinomial sampling. We also observe that coverage is higher for the GARW model that allows for time-varying growth advantage than for the FGA or MLR models which enforce a fixed growth advantage. As clades evolve and new subclades emerge we expect clade-specific growth advantage to change alongside.

In observing heterogeneity in prediction accuracy, we hypothesized that error is largest for emerging variants that present a small window of time to observe dynamics and where sequence count data is often rare. We investigate this hypothesis by charting how variant-specific growth advantage estimated in the MLR model varied across analysis dates (Fig 4). Generally, we see sharp changes in estimated growth advantage in the first 1–3 weeks when a variant is emerging, but then see less pronounced changes. Thus, it often takes several weeks for the MLR model to 'dial in' estimated growth advantage and accuracy will tend to be poorer in early weeks when variant-specific growth advantage is uncertain.

### Genomic surveillance systems and forecast error

Using the MLR model, we find that different countries have consistently different levels of forecasting error with forecasts in Brazil and South Africa showing more error than forecasts in the UK and the USA, while Trinidad and Tobago and Vietnam show more error than the other six countries (Fig 5A). We correlate broad statistics describing both quantity and quality of sequence data available in at different analysis time points and in different genomic surveillance systems to forecasting error (Fig 5B–5E). Using Pearson correlations we find that poor sequence quality as measured by proportion of available sequences labeled as 'bad' by Nextclade quality control [17] correlates slightly with mean AE (Fig 5B). We find that good sequence quantity as measured by total sequences available at analysis has a moderate negative correlation with mean absolute error (Fig 5E).

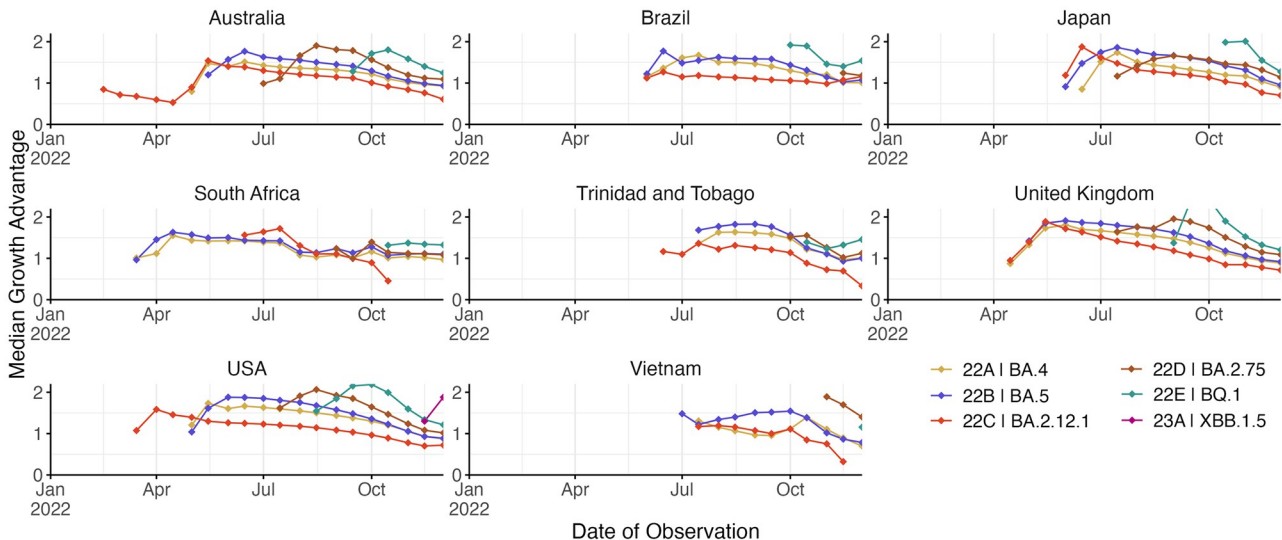

**Fig 4. Growth advantage of variants across analysis dates.** Growth advantage is estimated via the MLR model and is computed relative to clade 21K (lineage BA.1).

These results show that South Africa with ∼16k sequences collected in 2022 and median of 173 sequences available from the previous 30-days yields a mean absolute +30 day forecasting error of 7.0% for the MLR model (Table 1), which is only slightly greater than the mean absolute error of 6.3% for the US with ∼2.0M sequences collected in 2022 and of 5.8% for the UK with ∼1.2M sequences collected in 2022. However, Vietnam with ∼6k sequences collected in 2022 and median of 31 sequences available from the previous 30-delays yields a mean absolute forecasting error of 11.6% and Trinidad and Tobago with ∼2.3k sequences collected in 2022 and median of 44 sequences available from the previous 30-delays yields a mean absolute forecasting error of 12.0%. This suggests that genomic surveillance systems with cadence and throughput greater than 50–100 sequences collected in the previous 30 days yield sufficient timely data to permit short-term forecasts.

We follow up on this across-country analysis and subsample existing sequences from the United Kingdom and Denmark to investigate what number of sequences need to be collected weekly to keep forecast error within acceptable bounds. For context, we also computed the mean weekly sequences collected for selected countries globally in 2022 (Fig 6A). We select the United Kingdom due to its large counts of available sequences, relatively short submission delay, and low forecast error. Additionally, we include Denmark due to its large counts of available sequences and to explore the possibility of stochastic effects due to relative population sizes (Denmark has ∼9% the population of the UK). We simulate several downscaled data sets by subsampling the collected sequences at multiple thresholds for number of sequences per week and then fit the MLR model to each of the resulting data sets to see how forecast accuracy varies with sampling intensity. In order to properly account for variability in the subsampled data sets, we generate 5 subsamples per threshold, location and analysis date.

From this analysis, we find that increasing the number of sequences per week generally decreases the average error (Fig 6B and 6C), as well as decreasing the proportion of out-of-bounds predictions (Fig 6D and 6E), but there are diminishing returns. Additionally, the effect appears to saturate at different values depending on the forecast length. We find that for +14 and +30 day forecasts sampling at least 1000 sequences per week is fully sufficient to minimize

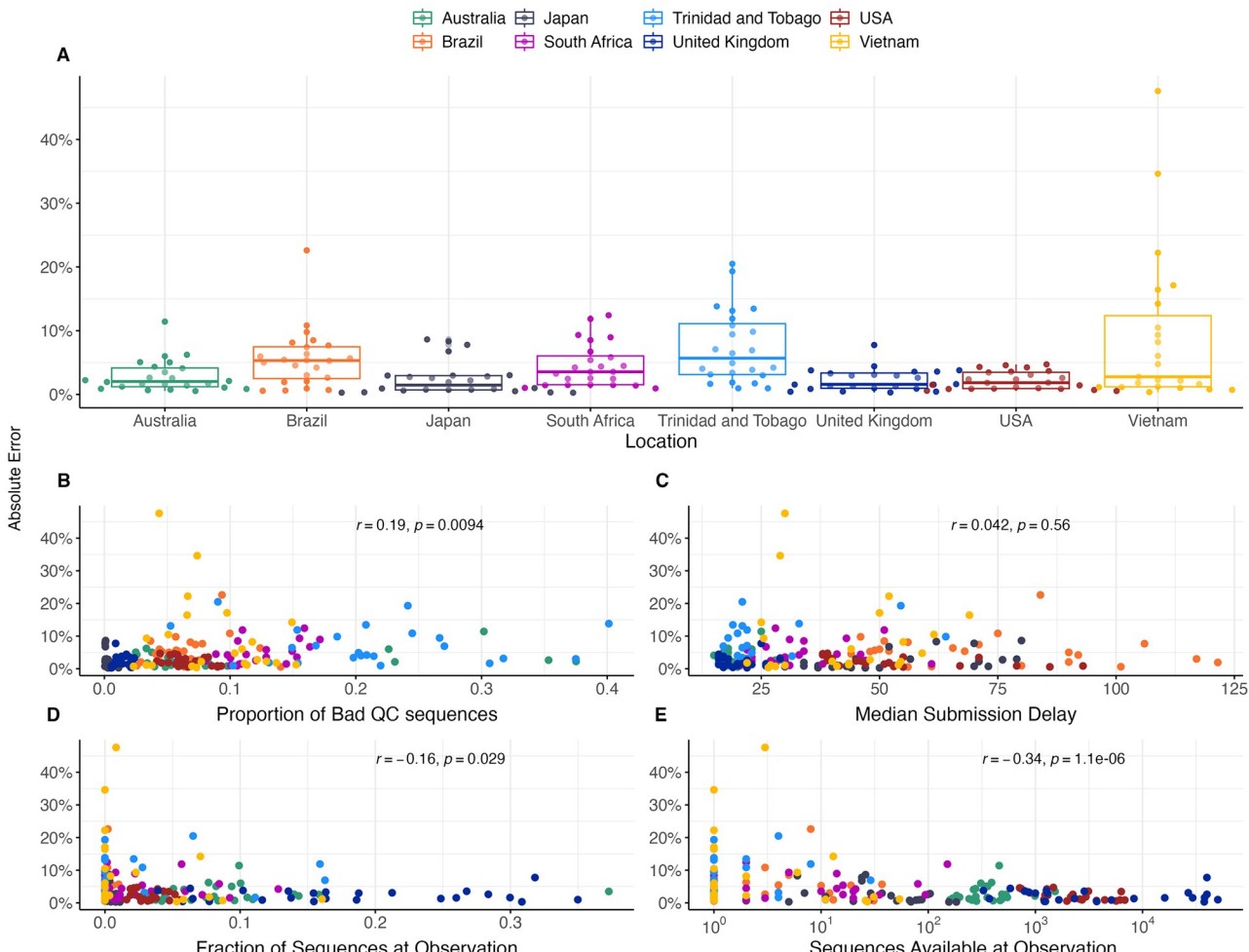

**Fig 5. Sequence quantity and quality influence nowcasts error.** (A) Absolute error at nowcast for the MLR model across countries. Points represent separate data sets at different analysis dates. Median and interquartile range of absolute errors are shown as box-and-whisker plots. (B-E) Correlation of sequence quality and sequence quantity metrics with absolute error. Points represent separate data sets at different analysis dates. Correlation strength and significance are calculated via Pearson correlation and are inset in each panel.

forecast error, and 200 sequences per week is largely sufficient to curtail error. We arrive at a similar threshold of 1000 sequences per week for both the UK and Denmark (Fig 6B–6E).

## Comparing country-level and hierarchical short-term forecast models

In observing poor performance in initial period of variant emergence (Fig 4), as well as poor performance in countries with less intensive genomic surveillance (Fig 5), we conclude lack of data results in poor fitness estimates and so poor predictive performance. Joint modeling of data from multiple countries has been proposed as a way to getting improved estimates of variant growth advantages in general and also specifically improving frequency estimates in low and middle income countries. Hierarchical or joint forecast models for short-term frequency forecasts typically operate by pooling parameters between 'groups' in a model. For our application, we pool the relative fitness of variants across countries, so that estimated relative fitnesses are informed by not just the observed relative fitness within a location, but also the relative fitnesses in other locations.

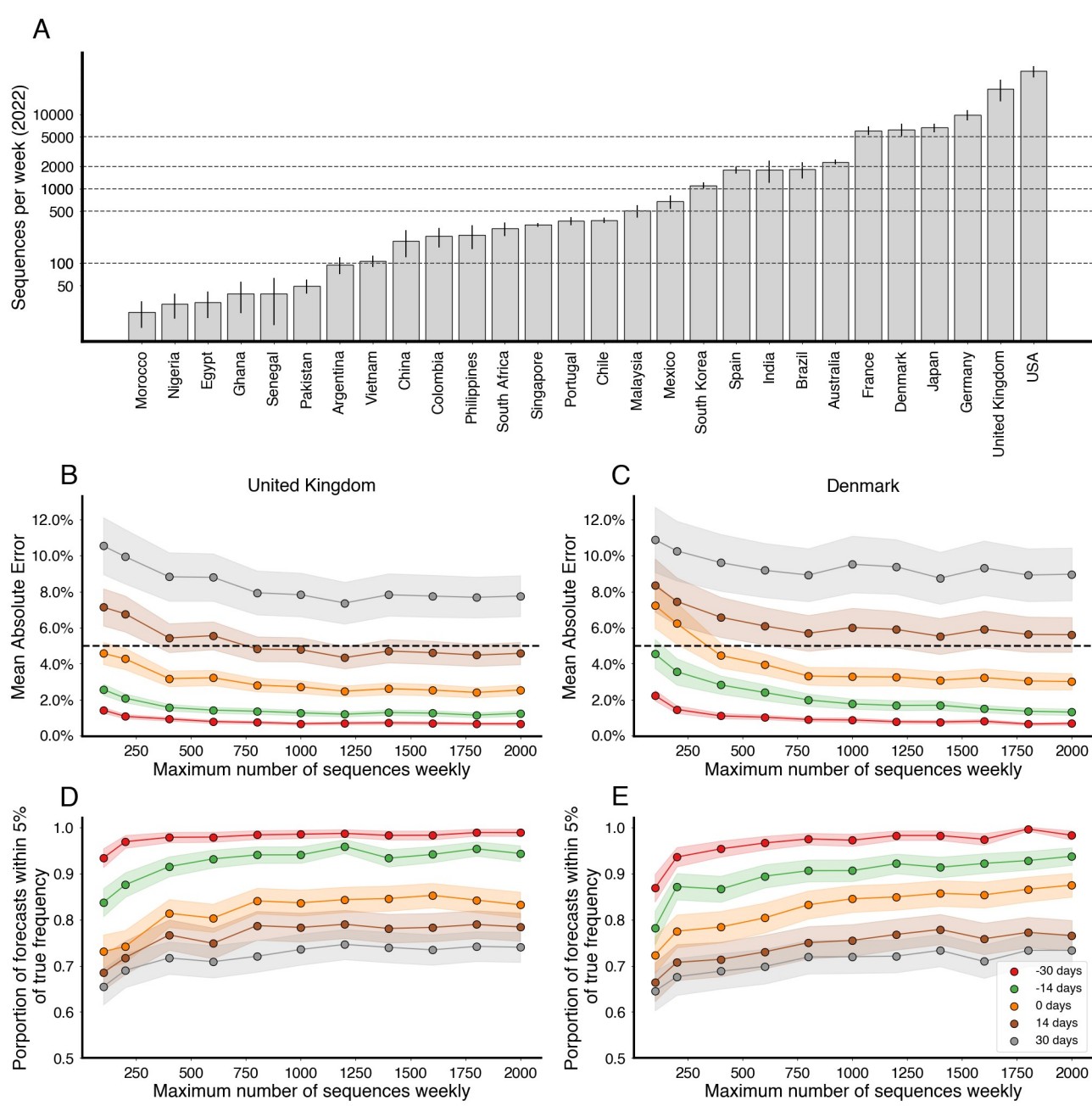

**Fig 6. Increasing sequencing intensity reduces forecast error.** (A) Mean sequences collected per week for selected countries in 2022. Intervals are 95% confidence intervals of the mean. Dashed lines correspond to sampling rates used in (B-E). (B, C) Mean absolute error as a function of sequences collected per week colored by forecast horizon (-30 days, -15 days, 0 days, +15 days, +30 days) for the United Kingdom and Denmark. The dash line corresponds to 5% frequency error. (D, E) Proportion of forecasts within 5% of retrospective frequency as a function of sequences collected for week for the United Kingdom and Denmark.

We compare the short-term forecast accuracy for individual models fit using MLR and this hierarchical MLR model in Fig 7. We find that overall the hierarchical MLR matches or outperforms the single country models in all locations and at all forecast lengths. Perhaps as expected the hierarchical MLR model matches MLR performance in countries with abundant

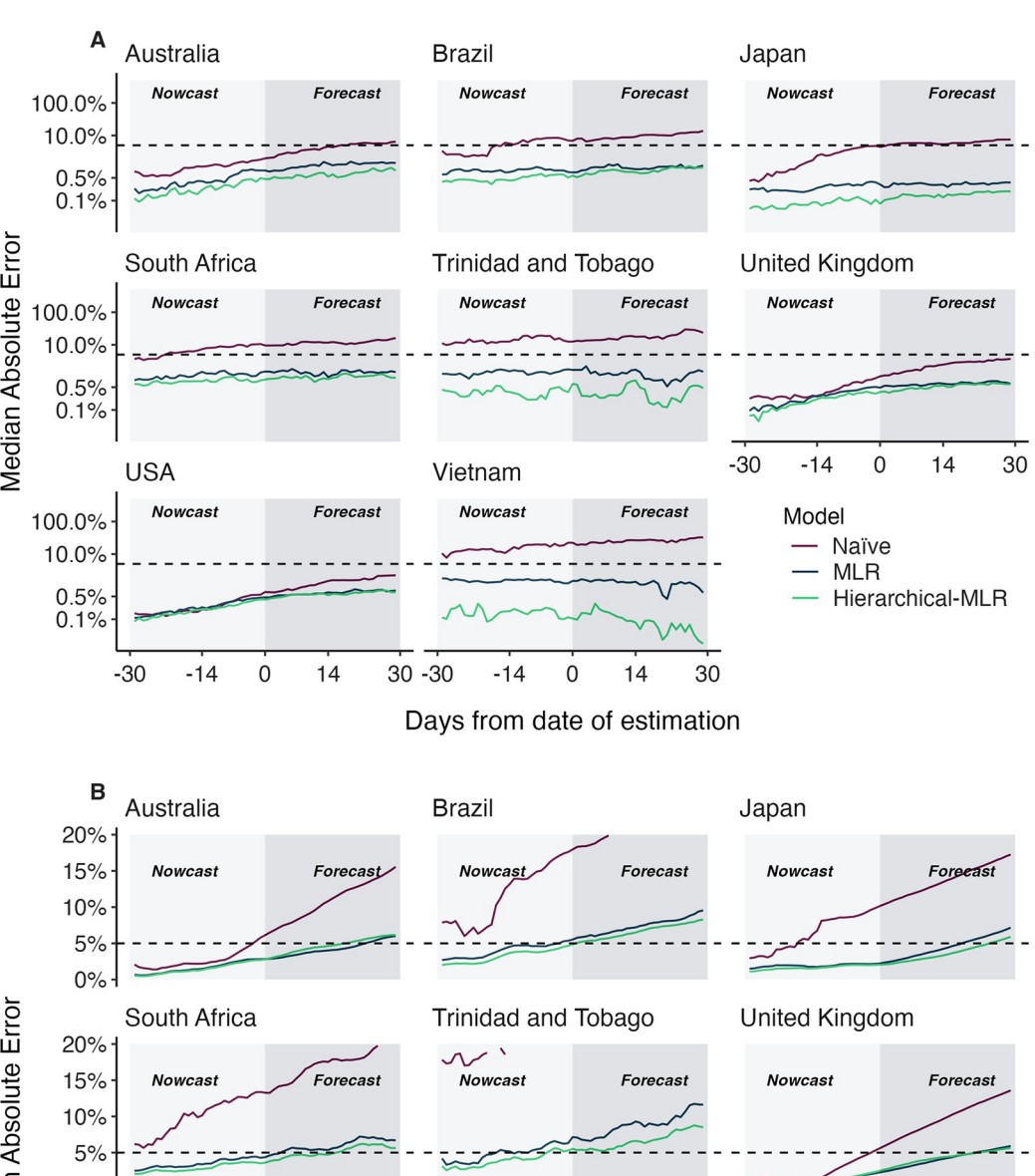

**Fig 7. Absolute error comparing standard MLR and hierarchical MLR across countries and forecast lags.** (A) Median absolute error and (B) mean absolute error across countries, models and forecast lags moving from −30 day hindcasts to +30 day forecasts. For each county / model / lag combination, the median and the mean are summarized across analysis data sets. Panel A uses a log y axis for legibility while panel B uses a natural y axis.

data like the US and UK, while countries with less data like Trinidad and Tobago and Vietnam show a large performance advantage to hierarchical MLR.

## Discussion

In this manuscript we sought to perform a comprehensive analysis of the accuracy of nowcasts and short-term forecasts from fitness models of SARS-CoV-2 variant frequency. We observe substantial differences between median and mean absolute error (Fig 2 and Table 1) with median errors generally quite well contained at 0.5–1.9% in the +30 day forecast, while mean errors are larger at 5.8–12.0%. This difference is due to the highly skewed distribution of model errors (Fig 3) where most predictions are highly accurate, but a smaller fraction are off-target. As expected, errors increase as target shifts from −30 day hindcast to +30 day forecast, but error increases more rapidly for mean absolute error than median absolute error. All four forecasting models explored here present a largely similar spectrum of errors.

We find that the Piantham, MLR, FGA and GARW models provide systematic and substantial improvements in forecasting accuracy relative to a 'naive' model that uses 7-day smoothed frequency at the last timepoint with sequence data (Fig 2 and Table 1). For the MLR model, at +30 days the improvement in median absolute error over naive is 1.4–31.0% and the improvement in mean absolute error is 6.8–26.2%. This result supports the use of MLR models in live dashboards like the CDC Variant Proportions nowcast (covid.cdc.gov/covid-data-tracker/#variant-proportions) and the Nextstrain SARS-CoV-2 Forecasts (nextstrain.org/sars-cov-2/forecasts/).

We also observe improvements in accuracy for the −30 day hindcast of modeled frequency relative to naive frequency with the MLR model showing improvement in median absolute error of 0.1–11.1% and improvement in mean absolute error of 0.9–17.0%. These improvements were greatest in countries with lower cadence and throughput of genomic surveillance (Trinidad and Tobago and Vietnam). Importantly, this suggests that fitness models are useful for hindcasts in addition to short-term forecasts and that −30 day retrospective frequency should not be taken as truth, ie it takes more time than 30 days for backfill to resolve retrospective frequency.

However, we observe that coverage is generally lower than ideal with predictive coverage under 50% for countries with the most sequencing (S8(B) Fig). We believe this may be due to a combination of over-dispersion of sequence counts relative to the multinomial sampling assumption as well as clade-level growth advantages changing through time as clades evolve. The former could be addressed by including over-dispersion in the sequence observation model and the latter could be addressed by implementing growth advantages that vary through time in an auto-correlated fashion.

We find that variability in forecast errors is partially driven by data limitations. When new variants are emerging, we lack sequence counts and lack time to observe growth dynamics resulting in initial uncertainty of variant growth rates (Fig 3). Relatedly, analyzing the variation in nowcast error, we find that overall sequence quality and quantity at time of analysis are associated with model accuracy (Fig 5). Thus, as expected, sequence quality, volume and turn-around time are all important for providing accurate, real-time estimates of variant fitness and frequency. Subsampling existing data in high sequencing intensity countries, we find that there are diminishing returns to increasing sequencing efforts and that maximum accuracy is achieved at around 1000 sequences per week and substantial accuracy is achieved at around 200 sequences per week (Fig 6). This level of sequencing enables robust short-term forecasts of pathogen frequency dynamics at the level of a country and highlights the feasibility of pathogen surveillance for evolutionary forecasting. As observed in Susswein et al.

susswein2023leveraging, pooling data across countries using a hierarchical fitness model improves short-term forecasts for SARS-CoV-2 variant dynamics (Fig 7).

In live MLR analyses at nextstrain.org/sars-cov-2/forecasts/, we have relied on the number of sequences available from samples collected in the previous 30-days as the key metric for inclusion of a country in the analysis. Along these lines, for pragmatic guidance for thresholds in which to trust MLR results, we observe that Trinidad and Tobago with 2.3k sequences collected in 2022 and a median 30-day sequence count of 43 shows a mean absolute forecasting error of 12%, that Vietnam with 6k sequences collected in 2022 and a median 30-day sequence count of 30 shows a mean absolute forecasting error of 11% and that South Africa with 16k sequences collected in 2022 and a median 30-day sequence count of 170 shows a mean absolute forecasting error of 7%. This suggests that a threshold of 50 sequences in previous 30 days should be roughly consistent with a $\sim 10\%$ forecasting error. Keeping forecasting error under 10% seems like a reasonable target for public display of frequency forecasts and would support targeting a threshold of 50 sequences from samples collected in the previous 30 days.

In addition to differences in genomic surveillance, we expect countries may differ in variant dynamics due to differences in absolute viral prevalence. We expect that variant frequencies we more closely follow the MLR expectation when absolute prevalence is large, achieved through a large host population and/or frequent repeated infection. Rapid continued evolution of SARS-CoV-2 [19] suggests that we will continue to see widespread circulation of SARS-CoV-2 and thus we generally expect fitness-based models to provide an adequate description of variant frequency dynamics. Of note, in comparing the UK with 67M people to Denmark with 6M people we observe similar levels of prediction error when sampling sequences at similar intensities (Fig 6). This suggests that stochastic effects from low absolute viral prevalence were not strongly manifesting with a population size of 6M. However, we do expect that at some smaller population size stochastic effects and repeated importations will cause a deviation in frequency dynamics from fitness model expectations.

Although these models appear largely accurate for short-term forecasts, they may be improved by incorporating underlying biological mechanism. In general, the methods discussed here are primarily statistical in nature and do not account for much of the biological or immunological knowledge that we have or could obtain. The incorporation of such knowledge could increase the short-term and medium-term capabilities of these models. Additionally, these fitness models do not account for future mutations and can only project forward from circulating viral diversity. This intrinsically limits the effective forecasting horizon achievable by these models. Future modeling work should seek to incorporate the emergence and spread of 'adjacent possible' mutations for longer term forecasts on the order of several months or years [20]. Without empirical frequency dynamics to draw upon, the fitness effects of these adjacent possible mutations may be estimated from empirical data such as deep mutational scanning [21–23]. Continued timely genomic surveillance and biological characterization along with further model development will be necessary for successful real-time evolutionary forecasting of SARS-CoV-2.

## Methods

### Preparing sequence counts and case counts

We prepared sequence count data sets to replicate a live forecasting environment using the Nextstrain-curated SARS-CoV-2 sequence metadata [24] which is created using the GISAID EpiCoV database [25]. To reconstruct available sequence data for a given analysis date, we filtered to all sequences with collection dates up to 90 days before the analysis date, and additionally filtered to those sequences which were submitted before the analysis date. These sequences

were tallied according to their annotated Nextstrain clade to produce sequence count for each country, for each clade and for each day over the period of interest. Sequence counts were produced independently for the 8 focal countries Australia, Brazil, Japan, South Africa, Trinidad and Tobago, the United Kingdom, the United States, and Vietnam. We repeated this process for a series of analysis dates on the 1st and 15th of each month starting with January 1, 2022 and ending with December 15, 2022 giving a total of 24 analysis data sets for each country. Since two models (FGA, GARW) also use case counts for their estimates, we additionally prepare data sets using case counts over the time periods of interest as available from Our World in Data (ourworldindata.org/covid-cases).

### Frequency dynamics and transmission advantages

We implemented and evaluated multiple models that forecast variant frequency. These models estimate the frequency $f_v(t)$ of variant $v$ at time $t$, and simultaneously estimate the variant transmission advantage $\Delta_v = \frac{R_t^v}{R_t^u}$ where $R_t^v$ is the effective reproduction number for variant $v$ and $u$ is an arbitrarily assigned reference variant with fixed fitness. We can interpret these transmission advantages as the effective reproduction number of a variant relative to some reference variant.

The four models of interest are: Multinomial Logistic Regression (MLR) of frequency growth, two models of variant-specific $R_t$: a fixed growth advantage model (FGA) parameterization and a growth advantage random walk (GARW) parameterization of the renewal equation framework of Figgins and Bedford [12], as well as another approach to estimating relative fitness by Piantham et al [13]. We provide a brief mathematical overview of these methods below.

The multinomial logistic regression model estimates a fixed growth advantage using logistic regression with a variant-specific intercept and time coefficient, so that the frequency of variant $v$ at time $t$ can be modeled as

$$f_v(t) = \frac{\exp(\alpha_v + \delta_v t)}{\sum_u \exp(\alpha_u + \delta_u t)},\tag{1}$$

where $\alpha_v$ is the initial frequency and $\delta_v$ is the growth rate of variant $v$, and the summation in the denominator is over variants 1 to $n$. Inferred frequency growth $f_v$ can be converted to a growth advantage (or selective coefficient) as $\Delta_v = \exp(\delta_v \tau)$ assuming a fixed deterministic generation time of $\tau$.

The model by Piantham et al [13] relies on an approximation to the renewal equation wherein new infections do not vary greatly over the generation time of the virus. This model generalizes the MLR model in that it accounts for non-fixed generation time though it assumes little overall case growth.

The fixed growth advantage (FGA) model uses a renewal equation model based on both case counts and sequence counts to estimate variant-specific $R_t$ assuming that the growth advantage $\Delta_v$ of variant $v$ is fixed relative to reference variant $u$ [12]. The growth advantage random walk (GARW) model uses the same renewal equation framework and data, but allows variant growth advantages to vary smoothly in time [12].

The models used all differ in the complexity of their assumptions in computing the variant growth advantage. Growth advantages presented in this manuscript are estimated relative to the initial Omicron strain (clade 21L, lineage BA.1), providing a point of reference for competing growth advantages and how median values change over time. Further details on the model formats can be found in their respective citations. All models were implemented using the

evofr software package for evolutionary forecasting (https://github.com/blab/evofr) using Numpyro for inference.

As a baseline, we compared the four models above to a naive model which generates the forecast as the average of the last available frequencies.

Additionally, we implement a hierarchical variant of the model where multiple countries are fit simultaneously with a Normal prior on the relative fitness of a given variant between countries, so that $\delta_{v,g} \sim \text{Normal}(\bar{\delta}_v, \sigma)$. Similar formulations of this hierarchical model have been used for SARS-CoV-2 frequency forecasts previously. [18]

## Evaluation criteria

We calculated the 'absolute error' (AE) for a given model $m$ and data set $d$ as the difference between the retrospective raw frequencies and the predicted frequencies as

$$\text{AE}_t^{m,d} = \frac{1}{n} \sum_{v \in V} \left| f_v^d(t) - \hat{f}_v^{m,d}(t) \right|, \tag{2}$$

where $f_v^d(t)$ and $\hat{f}_v^{m,d}(t)$ are the retrospective frequencies and the predicted frequencies for model $m$, data $d$, variant $v$ and time $t$. The AE is the mean across individual variants for a specific model, data set and time point. Additionally, we often work with the lead time which is defined as the difference between date of analysis for the data set and the forecast date $l = t - T_{\text{obs}}$. We summarized median absolute error and mean absolute error across multiple analysis datasets in Fig 2 and Table 1.

Throughout this study, we primarily use the median and mean absolute error to evaluate the accuracy of our point forecasts. We select the median absolute error as a measure of central tendency on our forecast errors, reducing the influence of outliers and skewed data distributions due to the contribution of forecasts which tend to diverge rapidly in forecast lead. To balance this and account for the effect of outliers and rapidly divergent forecasts, we also use the mean absolute error which is less sensitive to outliers than the mean square error and has units in terms of frequencies directly.

However, these are not the only possible choices for error metrics, and are motivated by our decision to focus primarily on point forecasts of variant frequencies. To supplement this analysis, we also address the coverage of probabilistic extensions of the models discussed here.

## Generating predictors of error

We explored four key variables to describe the effect of sequencing efforts on nowcast errors and estimated Pearson correlations with the mean absolute nowcast errors. These variables are defined as proportion of bad quality control (QC) sequences according to Nextclade [17], fraction of sequences available within 14 days of the prediction time, total sequences availability within 14 days of the prediction time and median delay of sequence submission. To calculate these variables, we selected a 14-day window of data before each and every analysis date and used the collection and submission dates to determine their availability. Total sequence availability was calculated by dividing the sequences where submission date was before the date of analysis by the total collected sequences and similarly fraction of sequences at observation was estimated. Sequence submission delay was calculated by taking the difference between the submission date and the date of collection. Bad QC sequence proportion was estimated by dividing the sequences with bad QC classification by the total collected sequences. Estimates were computed for all defined dates of analysis across all countries.

### Assessing coverage for short-term frequency forecasts

The main results of our analyses rely on mean and median absolute error as metrics, however, there is much to gain by using probabilistic forecasts for variant frequency. To this aim, we investigate the coverage of these different methods for forecasting variant frequency. Though not all models described initially were designed with uncertainty quantification in mind, we develop and fit Bayesian extensions of these models which are fit to the same data sets as before using stochastic variational inference.

### Downscaling historical sequencing effort

We analyze the effects of scaling back sequencing efforts to assess the effect of sequencing volume on nowcast and forecast errors. Using the sequencing data from the United Kingdom and Denmark, we subsampled existing available sequences at the time of analysis at a rate of 100, 200, 400, 600, 800, 1000, 1200, 1400, 1600, 1800, and 2000 sequences per week of any submission date. We then generated datasets for the same analysis dates and study period used in the previous analyses, generating 5 replicate subsampled data sets of sequences available at each analysis date for each eventual sequencing rate, location, and analysis date. Subsampling sequences per week before checking which sequences were available by the analysis date ensures that we respect the availability of sequences by submission date and submission delay in each country, so that countries with many sequences per week but long delays will maintain these delays. Therefore, the selected sequencing rate sets an upper limit on the number of sequences available per week at any analysis date and preserves the decline in available sequences that we typically observe in recent weeks since we only include sequences which are within our original subsample and available at the time of analysis. We then fit the MLR forecast model to each resulting data set and forecast up to 30 days after analysis date and compared these forecasts to the truth set in previous sections to compute the forecast error for each model. To better understand how the forecast error varies with sequencing intensity and forecast length, we computed the fraction of forecasts within an error tolerance (5% AE) as well as the average error at different sequence threshold and lag times.

### Comparing forecasts using retrospective clade designations and real-time designations

The main analyses discussed in this manuscript rely on subsetting and filtering SARS-CoV-2 sequence metadata accessed on a particular date. However, the clade designations used throughout this manuscript may not have been the same as clade designations at the time the data was available. To understand how this affects our evaluation of forecast error, we compare the accuracy of models fit to the sequence counts from metadata at the time and using the available Nextclade reference tree to those fit on the retrospective Nextclade reference tree used in the rest of the analyses in this paper. This compares lineage designations that were available in real-time on the historical analysis date to lineage designations that are retrospectively available. In particular, we focus on the timing of the designation of lineage BQ.1 (corresponding to clade 22E) in October 2022 and show the accuracy of MLR using the different data sets at different forecast leads. We compare the resulting MAE of these analyses between Nextclade versions in S9 Fig and show trajectories from individual countries in S10–S17 Figs.

## Supporting information

**S1 Fig. Reconstructing available data sets for Australia, Brazil, South Africa, Trinidad and Tobago, the United Kingdom, and Vietnam.** (A) Variant sequence counts categorized by

Nextstrain clade at 4 different analysis dates.
(TIF)

**S2 Fig. Reconstructing predictions for Australia.** (A) +30 day frequency forecasts for variants in bimonthly intervals using the MLR model for Australia. Each forecast trajectory is shown as a different colored line. Retrospective smoothed frequency is shown as a thick black line.
(TIF)

**S3 Fig. Reconstructing predictions for Brazil.** (A) +30 day frequency forecasts for variants in bimonthly intervals using the MLR model for Brazil. Each forecast trajectory is shown as a different colored line. Retrospective smoothed frequency is shown as a thick black line.
(TIF)

**S4 Fig. Reconstructing predictions for South Africa.** (A) +30 day frequency forecasts for variants in bimonthly intervals using the MLR model for South Africa. Each forecast trajectory is shown as a different colored line. Retrospective smoothed frequency is shown as a thick black line.
(TIF)

**S5 Fig. Reconstructing predictions for Trinidad and Tobago.** (A) +30 day frequency forecasts for variants in bimonthly intervals using the MLR model for Trinidad and Tobago. Each forecast trajectory is shown as a different colored line. Retrospective smoothed frequency is shown as a thick black line.
(TIF)

**S6 Fig. Reconstructing predictions for United Kingdom.** (A) +30 day frequency forecasts for variants in bimonthly intervals using the MLR model for United Kingdom. Each forecast trajectory is shown as a different colored line. Retrospective smoothed frequency is shown as a thick black line.
(TIF)

**S7 Fig. Reconstructing predictions for Vietnam.** (A) +30 day frequency forecasts for variants in bimonthly intervals using the MLR model for Vietnam. Each forecast trajectory is shown as a different colored line. Retrospective smoothed frequency is shown as a thick black line.
(TIF)

**S8 Fig. Posterior and predictive coverage for estimates across countries and models.** (A) The proportion of estimates lying within the 95% confidence intervals (CIs) of posterior latent frequencies across lag times (-30,-30). (B) The proportion of estimates lying within the 95% confidence intervals (CIs) of posterior predictive sample frequencies across lag times (-30,-30). We generate the posterior predictive sample frequencies by sampling random counts for each variant using their posterior latent frequencies conditioning on the total sequences being those observed retrospectively.
(TIF)

**S9 Fig. Comparing the accuracy of short-term forecast models under retrospective vs real-time clade assignments.** (A-H) Mean absolute error for MLR as a function of days since date of estimation, starting from 30 day hindcasts to 30 days forecasts. Intervals shown have width of two standard errors of the mean. We compare retrospective Nextstrain clade assignments made today ('Current Nextclade') to Nextstrain clade assignments available in Oct 2022 ('Real-time Nextclade'). We find that errors are qualitatively similar regardless of Nextclade version

with errors being potentially higher for the current Nextclade version.
(TIF)

**S10 Fig. Forecasts for Australia using clade designations under retrospective vs real-time clade assignments.** Forecasts from MLR fit to data generated using retrospective Nextstrain clade designations ('Current Nextclade') (A) and Nextstrain clade assignments available in Oct 2022 ('Real-time Nextclade') (B).
(TIF)

**S11 Fig. Forecasts for Brazil using clade designations under retrospective vs real-time clade assignments.** Forecasts from MLR fit to data generated using retrospective Nextstrain clade designations ('Current Nextclade') (A) and Nextstrain clade assignments available in Oct 2022 ('Real-time Nextclade') (B).
(TIF)

**S12 Fig. Forecasts for Japan using clade designations under retrospective vs real-time clade assignments.** Forecasts from MLR fit to data generated using retrospective Nextstrain clade designations ('Current Nextclade') (A) and Nextstrain clade assignments available in Oct 2022 ('Real-time Nextclade') (B).
(TIF)

**S13 Fig. Forecasts for South Africa using clade designations under retrospective vs real-time clade assignments.** Forecasts from MLR fit to data generated using retrospective Nextstrain clade designations ('Current Nextclade') (A) and Nextstrain clade assignments available in Oct 2022 ('Real-time Nextclade') (B).
(TIF)

**S14 Fig. Forecasts for Trinidad and Tobago using clade designations under retrospective vs real-time clade assignments.** Forecasts from MLR fit to data generated using retrospective Nextstrain clade designations ('Current Nextclade') (A) and Nextstrain clade assign- ments available in Oct 2022 ('Real-time Nextclade') (B).
(TIF)

**S15 Fig. Forecasts for United States using clade designations under retrospective vs real-time clade assignments.** Forecasts from MLR fit to data generated using retrospective Nextstrain clade designations ('Current Nextclade') (A) and Nextstrain clade assignments available in Oct 2022 ('Real-time Nextclade') (B).
(TIF)

**S16 Fig. Forecasts for United Kingdom using clade designations under retrospective vs real-time clade assignments.** Forecasts from MLR fit to data generated using retrospective Nextstrain clade designations ('Current Nextclade') (A) and Nextstrain clade assignments available in Oct 2022 ('Real-time Nextclade') (B).
(TIF)

**S17 Fig. Forecasts for Vietnam using clade designations under retrospective vs real-time clade assignments.** Forecasts from MLR fit to data generated using retrospective Nextstrain clade designations ('Current Nextclade') (A) and Nextstrain clade assignments available in Oct 2022 ('Real-time Nextclade') (B).
(TIF)

## Acknowledgments

We thank John Huddleston for many helpful comments on the approach and on the manuscript. We gratefully acknowledge all data contributors, ie the Authors and their Originating laboratories responsible for obtaining the specimens, and their Submitting laboratories for generating the genetic sequence and metadata and sharing via the GISAID Initiative, on which this research is based. We have included an acknowledgements table in the associated GitHub repository under `data/final_acknowledgements_gisaid.tsv.gz`.

## Author Contributions

**Conceptualization:** Eslam Abousamra, Marlin Figgins, Trevor Bedford.

**Data curation:** Eslam Abousamra, Marlin Figgins.

**Formal analysis:** Eslam Abousamra, Marlin Figgins.

**Funding acquisition:** Trevor Bedford.

**Investigation:** Trevor Bedford.

**Methodology:** Marlin Figgins.

**Project administration:** Eslam Abousamra, Trevor Bedford.

**Resources:** Trevor Bedford.

**Software:** Marlin Figgins.

**Supervision:** Trevor Bedford.

**Visualization:** Eslam Abousamra, Marlin Figgins.

**Writing – original draft:** Eslam Abousamra.

**Writing – review & editing:** Eslam Abousamra, Marlin Figgins, Trevor Bedford.

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
