## [Decision Letter · Decision Letter 0]

23 Feb 2024

Dear Mr Abousamra,

Thank you very much for submitting your manuscript "Fitness models provide accurate short-term forecasts of SARS-CoV-2 variant frequency" for consideration at PLOS Computational Biology.

As with all papers reviewed by the journal, your manuscript was reviewed by members of the editorial board and by several independent reviewers. In light of the reviews (below this email), we would like to invite the resubmission of a significantly-revised version that takes into account the reviewers' comments.

I agree with the reviewers that this paper represents a potentially impactful contribution related to SARS-CoV-2 variant fitness estimation, which is key for accurate risk assessment and time series forecasting. However, I also agree strongly with R1 that the authors need to account for the unstable nature of lineage assignment as new variants emerge. I'm less familiar with the operational specifics of nextstrain's assignment, but with pangolin is should be possible to use the model version in use during each variant emergence period. I also agree with both reviewers that more effort is needed on uncertainty quantification and on measuring performance in geographic locations with lower genome sampling effort. I hope that the authors pay careful attention to the detailed reviewer comments, as following them will lead to a meaningful improvement in the manuscript.

We cannot make any decision about publication until we have seen the revised manuscript and your response to the reviewers' comments. Your revised manuscript is also likely to be sent to reviewers for further evaluation.

Sincerely,

Samuel V. Scarpino

Academic Editor

PLOS Computational Biology

Thomas Leitner

Section Editor

PLOS Computational Biology

I agree with the reviewers that this paper represents a potentially impactful contribution related to SARS-CoV-2 variant fitness estimation, which is key for accurate risk assessment and time series forecasting. However, I also agree strongly with R1 that the authors need to account for the unstable nature of lineage assignment as new variants emerge. I'm less familiar with the operational specifics of nextstrain's assignment, but with pangolin is should be possible to use the model version in use during each variant emergence period. I also agree with both reviewers that more effort is needed on uncertainty quantification and on measuring performance in geographic locations with lower genome sampling effort. I hope that the authors pay careful attention to the detailed reviewer comments, as following them will lead to a meaningful improvement in the manuscript.

Reviewer's Responses to Questions

**Comments to the Authors:**

Reviewer #1: Reivew is uploaded as an attachment

Reviewer #2: It was a joy to read this ms. The authors propose a framework to evaluate real-time forecasts of variant frequencies. They use the evolution of SARS-CoV-2 in 2022 as a case study.

They use their framework to compare models across different countries on a range of genomic surveillance intensity.

Perhaps what I was most excited about was the use of it to estimate minimum number of sequences needed per week is sufficient for accurate short-term forecasts. This is an incredible relevant question not just from a methodological standpoint but of capacity building and public health importance.

I have a few minor suggestions:

I suggest you use countries with much lower genomic surveillance effort as well, as while there are some differences between those countries they are all on higher end of effort. Meaning that the issues especially at the emergence of a new variant will be easier to miss.

It wasn’t clear why for Australia model’s performance was slightly different. Is it due to sampling efforts or it spurious? I suggest the authors explain this a little in more detail.

Not surprisingly, the authors found that variability in forecast errors is partially driven by data limitations. And thus, as expected, the authors focus on hypothesis that error is largest for emerging variants that present a small window of time to observe dynamics and where sequence count data is often rare. So, while their testing is sound I suggest two additional steps:

1. This would no doubt be exacerbated in countries with low effort or very pulsed effort. It would be great to see that explored. So picking two or three countries with constantly low effort and or pulsed effort will be very useful

2. To this regard, I would suggest to the authors to take their framework to evaluate models that are specifically pooling country data, such as https://www.medrxiv.org/content/10.1101/2023.01.02.23284123v4.full (cited in the ms) which address a lot of the issues related with backfilling and the under/ over estimation at emergence, heterogenous reporting. These pooled models while MLR, should be evaluated separately from single country models, for all the reasons and issues the authors state they are exploring.

While I agree with the statements that MLR models, “do not account for future mutations and can only project forwards from circulating viral diversity. This intrinsically limits the effective forecasting horizon achievable by these models.” It seems to me that the discussion should center in the purpose of the models to begin with. So, for a more balanced discussion, I would suggest the authors to add perhaps a paragraph in the discussion on the role of adding mechanism or background information (vaccine, previous waves, to understand immunity profiles of populations) onto these models, if they are to be useful beyond the nowcast, as it will no doubt affect internal dynamics and competition between variants that in turn affects sustained growth and displacement of other less fit variants.

**Have the authors made all data and (if applicable) computational code underlying the findings in their manuscript fully available?**

Reviewer #1: Yes

Reviewer #2: Yes

PLOS authors have the option to publish the peer review history of their article (what does this mean?). If published, this will include your full peer review and any attached files.

Reviewer #1: No

Reviewer #2: No
---

## [Decision Letter · Decision Letter 1]

29 Jul 2024

Dear Mr Abousamra,

Thank you very much for submitting your manuscript "Fitness models provide accurate short-term forecasts of SARS-CoV-2 variant frequency" for consideration at PLOS Computational Biology. As with all papers reviewed by the journal, your manuscript was reviewed by members of the editorial board and by several independent reviewers. The reviewers appreciated the attention to an important topic. Based on the reviews, we are likely to accept this manuscript for publication, providing that you modify the manuscript according to the review recommendations.

Sincerely,

Samuel V. Scarpino

Academic Editor

PLOS Computational Biology

Thomas Leitner

Section Editor

PLOS Computational Biology

Reviewer's Responses to Questions

**Comments to the Authors:**

Reviewer #1: Overall, the manuscript is of great importance to public health and specifically estimation and evaluation of variant dynamics. The authors made significant improvements to the manuscript in the course of their revisions, and I would highly recommend this article for publication in PLOS Computational Biology, contingent on a few minor revisions/suggestions:

Revisions:

- Suggested edit in methods titled “Comparing forecasts using retrospective clade designations and real-time designations” -> “The main analysis discussed in…. accessed on a particular date” (date instead of what is written now which is data)

- Suggest clarifying terminology, when referring to Nextclade version to indicate that this means the clade designation model. It might be confusing to a reader who is already thinking about time-stamped versions of the data to distinguish between the two, but this distinction is critical to understanding the difference between the real-time and retrospective variant calling.

- I would suggest calling a variant by its pangolin with Nextclade beside it throughout e.g. BQ.1 (Omicron 22E)

Suggestions:

- I would clarify in the description of the downsampling, that we might expect that when sequencing declines overall, it would also decline in recent weeks, instead of the recent weeks being the upper limit, and so the analysis may not reflect the real-time impact of reducing sequencing quantity unless the speed of sequencing is prioritized.

- I’m not sure why the hierarchical MLR results are not included in Figures 2 and 3, and instead are an add on in Figure 7. The manuscript might read more smoothly if the hierarchical MLR is presented alongside the other models including the single country MLR. Right now it appears like a secondary analysis

- I would suggest that the authors describe in further detail why they think the coverage is so low in high sequencing countries? Is this because the model is underdispersed/over confident? If so, could they offer any possible solutions to this problem in the discussion (e.g. a different observation model, models accounting for heterogeneity in facilities, etc.)

- Figures S10-17 are a bit confusing, which lines represent the fit to the real time next clade and which lines are fit to the current nextclade? As presented, there appears to only be one model fit, it would be helpful to present the model fit to both input data sets to compare accuracy to both the real time and retrospective assignments. In general, like these figures though because they make clear some examples of the forecasts versus the later observations.

Reviewer #3: In the article: "Fitness models provide accurate short-term forecasts of SARS-CoV-2 variant frequency" the authors provide a nice demonstration of the performance of different variant forecasting models for SARS-CoV-2 variants and provide estimates for the sample size necessary for countries surveillance programs. The article is well written and covers an important topic, particularly as the US is considering a new variant forecasting competition. Based on their revisions to reviewer feedback I believe this article should be accepted for publication, but I have a single suggestion the authors should consider adding.

Suggestion: The authors provide a sample size that they feel would be sufficient to have reasonable data for stable variant forecasting based on a nice downsampling analysis. One issue is that many countries are not currently reporting 1,000 cases a week, some not even 200! It would be useful if the authors could try and speak to these countries regarding the genomic sequence sample sizes necessary. My feeling is that it should be reasonable to carry out nowcast/forecasts in a scenario where there are 200 reported cases a week and all of them are sequenced, though I think it's an interesting question. It would be great if the authors could analyze the fraction of reported cases being sequenced in their sample size analysis (maybe some simple back-of-the-envelope calculations could be made) to see if they can provide any guidance in these scenarios where there are smaller reported case counts.

**Have the authors made all data and (if applicable) computational code underlying the findings in their manuscript fully available?**

Reviewer #1: Yes

Reviewer #3: Yes

PLOS authors have the option to publish the peer review history of their article (what does this mean?). If published, this will include your full peer review and any attached files.

Reviewer #1: No

Reviewer #3: No

Figure Files:

Data Requirements:

Reproducibility:

References:

---

## [Editor Report · Decision Letter 2]

28 Aug 2024

Dear Mr Abousamra,

We are pleased to inform you that your manuscript 'Fitness models provide accurate short-term forecasts of SARS-CoV-2 variant frequency' has been provisionally accepted for publication in PLOS Computational Biology.

Best regards,

Samuel V. Scarpino

Academic Editor

PLOS Computational Biology

Thomas Leitner

Section Editor

PLOS Computational Biology

---

## [Editor Report · Acceptance letter]

4 Sep 2024

PCOMPBIOL-D-23-01974R2 

Fitness models provide accurate short-term forecasts of SARS-CoV-2 variant frequency

Dear Dr Abousamra,

I am pleased to inform you that your manuscript has been formally accepted for publication in PLOS Computational Biology. Your manuscript is now with our production department and you will be notified of the publication date in due course.

With kind regards,

Zsofia Freund
